# Topical Omega-3 Fatty Acids Eyedrops in the Treatment of Dry Eye and Ocular Surface Disease: A Systematic Review

**DOI:** 10.3390/ijms232113156

**Published:** 2022-10-29

**Authors:** Benjamin Paik, Louis Tong

**Affiliations:** 1Lee Kong Chian School of Medicine, Nanyang Technological University, Singapore 308232, Singapore; 2Department of Cornea and External Eye Disease, Singapore National Eye Center, Singapore 168751, Singapore; 3Ocular Surface Research Group, Singapore Eye Research Institute, Singapore 169856, Singapore; 4Ophthalmlogy and Visual Sciences Academic Clinical Programme, Duke-NUS Medical School, Singapore 169857, Singapore

**Keywords:** dry eye, ocular surface, inflammation, omega-3

## Abstract

Dry eye is a common inflammatory condition of the ocular surface. While oral omega-3 supplementation for its treatment has been extensively studied, recent large-scale studies have cast doubt on their efficacy. However, efficacy of topical omega-3 has yet to be reviewed. We performed a systematic search of PubMed, Embase, and Cochrane databases for all studies evaluating topical omega-3 in dry eye. Five human and five animal studies were included. Of the five human studies, two were on dry eye disease (DED), one was on contact lens discomfort, and two were on patients undergoing corneal collagen crosslinking. In humans, there is promising evidence for improved ocular surface staining and tear break-up time compared to controls, equivocal evidence for improvements to ocular surface symptoms and meibomian gland dysfunction, and no effect on increasing tear production. Data from animal models largely agree with these findings, and further reveal decreased inflammatory cytokines and monocyte infiltration. Our review suggests that topical omega-3 is a promising treatment for dry eye, but also points to the paucity of evidence in this field. Further trials in humans are required to characterize effects of topical omega-3 and optimize its dosage.

## 1. Introduction

Dry eye disease (DED) is a chronic condition affecting up to 50% of the population [1] and is a common reason for seeking ophthalmic care [2]. Frequent outpatient visits and loss of productivity from symptoms leads to a significant economic burden [3]. DED is a broad “umbrella” term for inflammation of the ocular surface (cornea, conjunctiva, lacrimal apparatus, eyelids) [4] featuring tear film dysfunction, and can arise on a background of nonspecific risk factors (any cause of meibomian gland dysfunction (MGD), conjunctivitis, or blepharitis) and/or a specific cause (such as contact lens usage [5,6], ocular surgery [7]). In health, the ocular surface is coated with an even layer of tear film which protects the cornea from stressors including dessication, environmental pathogens, and allergens [8]. However, in ocular surface disease such as DED, there are abnormalities in tear film quantity and quality [9,10,11]. The normal tear film is thought to consist of three components [12]; dysfunction exposes the ocular surface to stressors and incites localized inflammation, leading to a chronically persistent “vicious cycle” of further ocular surface damage, worsened inflammation, and deteriorating tear film function [13,14,15].

Treatment of DED aims to restore tear film function through supplementation with artificial tears, addressing risk factors if present, and underlying systemic disease if active [16]. Treatment also aims to dampen ocular inflammation with anti-inflammatory eyedrops such as corticosteroids or cyclosporine [17]. However, both treatments have undesirable side effects: prolonged steroid usage risks glaucoma, whereas cyclosporine causes a burning sensation [18], leading to low patient compliance. Therefore, while ocular inflammation is present across all severities and etiologies of DED [15], patients with mild to moderate DED can potentially benefit from methods to decrease ocular surface inflammation. There has been a rising interest in nutritional supplements as alternatives to pharmacological treatments to perform this role [19]. One such nutritional supplement, omega-3 (ω-3) fatty acids, dates back to the 18th century when cod liver oil was used in the Manchester Infirmary [20]. Key compounds in cod liver oil responsible for the anti-inflammatory properties include eicosapentaenoic acid (EPA), docosahexaenoic acid (DHA), and alpha-linoleic acid (ALA), together referred to as the ω-3 fatty acids (ω-3 PUFAs) [21]. Together with omega-6 (ω-6) PUFAs such as linoleic acid (LA) and arachidonic acid (AA), these compounds are obtained from the diet, absorbed in the gastrointestinal tract, and form a repertoire of lipid mediators (eicosanoids) which regulate inflammation by acting on innate and adaptive immune cells [22,23]. While ω-6 produces potent pro-inflammatory metabolites, supplementation with ω-3 produces lipid mediators which have an overall pro-resolving effect on inflammation [24,25]. Recent evidence has elucidated other metabolites exclusively derived from ω-3 such as resolvins, protectins, and maresins, which promote resolution of inflammation [26]. Therefore, studies point towards an optimal ω-6:ω-3 ratio of <4:1 and its optimization as key in dampening inflammation [27,28].

However, not all studies report positive results. A recent multicenter randomized trial, the DREAM study [29], found no added benefit of oral ω-3 over placebo, and results from the DREAM extension study [30] showed that patients who stopped ω-3 supplementation were not worse off than their counterparts who continued on ω-3.

Alternate routes of administration of ω-3, such as topically as an eyedrop, have been proposed. The postulated advantage of topical ω-3 lies in a greater ocular bioavailability [31]. Topical ω-3 ameliorates the need for absorption of ω-3 through the gastrointestinal tract [32]. Furthermore, studies on DED patients’ tears demonstrate higher ω-6:ω-3 ratios in the tear film correlating with worsened MGD and corneal staining, suggesting room for direct ω-3 supplementation into tear film [33]. However, topical eyedrops are plagued by tear washout and dilution in the tear, among other factors [34]. The actual efficacy of topical ω-3 needs to be evaluated in DED studies.

Therefore, we assembled real-world evidence in a systematic review of topical ω-3 in DED, to summarize its outcomes of treatment on ocular surface inflammatory diseases, and to shed light on its clinical efficacy and safety.

## 2. Methods

This systematic review was performed in accordance with the Preferred Reporting Items for Systematic Reviews and Meta-Analyses (PRISMA) statement guidelines [35].

### 2.1. Search Strategy

An electronic search was performed using MEDLINE (via PubMed), EMBASE (via Ovid), and the Cochrane Library from inception till 16 October 2022 with no language restriction, using a combination of both MeSH and non-MeSH keywords: “ω-3” (MeSH) and “topical or eyedrop” and “eye” (MeSH) and modified the search syntax for compatibility. All possible relevant articles were included by performing backward reference searching of included articles and review articles.

### 2.2. Eligibility Criteria

We included all articles which reported outcomes of topical ω-3 in the form of eyedrop formulations in the treatment of DED or these conditions which feature ocular surface inflammation (e.g., contact lens discomfort, corneal collagen crosslinking (CXL)), but excluded all conditions of a predominantly infectious etiology (microbial keratitis, viral conjunctivitis, etc.). We also included animal studies that reported the effects of topical ω-3 given to the ocular surface. However, we excluded all cell and tissue-based studies, technical notes, revies, and studies which administered biologically active metabolites (e.g., PEDF [36] and resolvins [37,38]) because these compounds are expected to boost the anti-inflammatory response by a disproportionate extent compared to ω-3 alone, and hence skew the results. We also included conference abstracts or letters from PubMed if there was sufficient meaningful data to be extracted and analyzed. If there were articles with overlapping patient populations (same study group, identical population size and interventions) we chose the publication with the larger sample size or the study that provided the most pertinent clinical information.

### 2.3. Selection of Studies and Data Extraction

Following a comprehensive search and deduplication, we initially screened all articles by title and abstract only. The full texts of articles included in the first stage were subsequently reviewed in their entirety for final inclusion. A standardized data sheet was used to extract our outcome measures from text, tables, and figures in animal and human studies. All human studies were assessed for risk of bias using the National Institute of Health Quality Assessment Tools [39].

### 2.4. Outcome Measures

Across all studies, we extracted the following data if available: data on symptoms of ocular surface disease measured using the Ocular Surface Disease Index (OSDI) and Contact Lens Dry Eye Questionnaire-8 (CLDEQ-8) scores; data on other slit-lamp examination findings; data on tear function parameters (tear breakup time (TBUT), Schirmer’s, corneal and conjunctival staining scores, tear meniscus height, corneal nerve fiber density); biochemical data on cytokine elaboration and ocular surface immune cell enumeration.

### 2.5. Data Analysis and Quality Assessment (Risk of Bias)

The authors elected to perform a descriptive review of the data as opposed to a meta-analysis due to the heterogeneity of the studies assessed. Risk of bias was assessed using the National Institute of Health (NIH) Quality Assessment Tool for Controlled Intervention Studies, or the NIH Tool for Before–After (Pre–Post) studies with no control group (Appendix A).

## 3. Results and Discussion

### 3.1. Systematic Search Strategy

The initial search revealed a total of 152 publications for possible inclusion after manual deduplication, of which 16 remained for full-text review after title and abstract screening. From the remaining 16 publications, six were excluded after applying our exclusion criteria, as shown in the PRISMA diagram (Figure 1). This left 10 publications for the final analysis, of which 5 were human studies (Table 1) and 5 were animal studies (Table 2).

### 3.2. Human Studies

#### 3.2.1. Study Design

A total of five articles [40,41,42,43,44] investigating the efficacy of topical ω-3 eyedrop formulations in the treatment of ocular surface disease were identified, of which two were on idiopathic DED [40,43], one was on DED due to contact lens discomfort (CLD-DED) [44], and two were on DED secondary to corneal collagen cross-linking (CXL) surgery (CXL-DED) [41,42].

A total of 419 patients were included. All articles were parallel studies of a prospective nature, monitored patients at predetermined intervals, and collected data on signs and symptoms pre- and post-treatment. Among the double-arm studies, most used artificial tears [43], hyaluronic acid (HA) [41,42] or surfactant F_6_H_8_ [40] as control. The treatment regimen ranged from 2 to 4 times a day for 2 to 4 months. The vehicle composition of all eyedrops varied widely and included a combination of surfactants, lubricants, osmoprotectants, and antioxidants. Apart from one study [40] which mandated lid hygiene throughout, other studies did not introduce new concomitant treatments; the concomitant treatment(s) or lack thereof could be inferred from study-specific inclusion and exclusion criteria.

Two studies [40,43] contributed 273 patients with idiopathic DED, comprising a prospective observational (PO) single-arm open-label study on DHA-containing eyedrops by Jacobi [40], and a double-masked RCT by Downie [43] comparing a flaxseed oil-containing eyedrop with Refresh Optive Advanced. While Jacobi only included patients with DED due to MGD, Downie included all DED patients. Both studies included patients of older age groups (mean age > 50) who had moderate [43] to severe [40] DED, had a female preponderance of around 70% in both studies, and excluded patients using contact lenses, desiccating eyedrops, and with a history of corneal trauma; of note, only Downie’s study allowed patients to continue any cyclosporine eyedrops during their study. Another study on CLD-DED patients compared topical or oral EPA/DHA to olive oil control [44]. All patients were soft lens wearers for >6 months with symptomatic CLD and minimal to mild corneal staining.

Another two RCTs contributing 90 patients were on CXL-DED in young keratoconus patients (age: 22–37 years) [41,42]. The CXL procedure causes dysfunctional tear film due to the inevitable damage to the corneal subepithelial nerve plexus during the procedure which is essential for normal tear film secretion. In each study, all patients received the same postoperative antibiotics and steroids.

#### 3.2.2. Improvements in Ocular Symptoms

The results of the outcome measures of human studies are included in Table 3.

Ocular symptoms can be quantified and charted over time with the Ocular Surface Disease Index (OSDI) or Contact Lens Dry Eye Questionnaire (CLDEQ-8), which qualify the severity of ocular irritation.

Topical ω-3 improved OSDI from baseline and compared to control among idiopathic DED patients in 1 [40] out of 2 studies [40,43]. There was no improvement to CLDEQ-8 among CLD-DED patients [44].

In patients post-CXL, topical ω-3 for 3 months restored OSDI back to preoperative (healthy) values compared to HA control, at the 3-month timepoint. Interestingly, after treatment cessation, both control and ω-3 groups were restored to preoperative values by the 6-month timepoint, which is the time corneal nerves take to regenerate (range 6–12 months) [50].3.2.3. Improvements in Ocular Surface (Corneal and Conjunctival) Staining

Various ocular staining methods are used to highlight the areas of ocular surface damage in ocular surface inflammation. For instance, sodium fluorescein (NaFl) stains cornea areas corresponding to regions with loose or desquamated epithelial cells. Severity of staining in each of the five zones of the cornea is graded on a numerical scale, and summed up to obtain the total corneal fluorescein staining score (tCFS). Similarly, conjunctival staining is assessed using lissamine green (LG), which stains damaged conjunctival epithelium unprotected by the overlying mucin layer they usually secrete.

Topical ω-3 reduced tCFS in DED [40,43] from baseline and compared to controls, but not in CLD [44]. Yilmaz’s study [41] showed that patients receiving topical ω-3 post-CXL had lower tCFS compared to HA control at 1 month postoperatively (*p* = 0.012).

LG was used to assess conjunctival staining in two studies [43,44], of which one on DED patients [43] reported significant benefit of topical ω-3 over control and relative to baseline in DED patients. The other study on CXL patients found no benefit [44].

#### 3.2.3. Improvements in Tear Stability

Tear stability is essential to normal tear function, and is measured most frequently by TBUT, which is the time interval elapsed between a complete blink and the first appearance of a break in the tear film [51]. Tear stability is a part of normal tear film function, and any quantitative or qualitative tear deficiency [15] can lead to a shortened TBUT, but is most commonly due to tear mucin deficiency (conjunctivitis) or tear lipid deficiency (MGD) [52].

Among patients with chronic dry eye, ω-3 improved TBUT in one out of two studies on idiopathic DED [40,43] but not in CLD-DED [44]. In patients who underwent CXL, postoperative topical ω-3 actually increased the postoperative TBUT [41]. This is surprising because CXL destroys corneal epithelium and hence decreases TBUT [53], as manifested in the control group receiving HA.

#### 3.2.4. Reduction of Meibomian Dysfunction (MGD)

Normal function of meibomian glands is contingent on the unobstructed flow of meibum. Regular blinking forces meibum out of the glands and prevents their inspissation (and eventual obstruction) in MGD. The color and consistency of secretions (or lack thereof) from each expressed gland is scored on a scale of 0 to 3 (0 = transparent fluid; 1 = white and thick; 2 = toothpaste-like secretions; 3 = no expression). Five glands are typically assessed, and their cumulative score forms the MGD score (0–15) [54].

Across two studies which assessed MGD status pre- and post-treatment with topical ω-3, one DED study [40] reported improvement in MGD score from baseline, while another study on CLD patients [44] reported no advantage of topical ω-3 over placebo (oral olive oil).

#### 3.2.5. Tear Production and Volume

To assess tear production, three studies performed variations of the Schirmer’s test. A variant, Schirmer’s II, pre-treats the eye with topical anesthetic to eliminate reflex tearing and only assesses tearing from the accessory lacrimal glands, which control basal tear levels [51]. Two studies [41,44] performed Schirmer’s II while one study [40] performed Schirmer’s I. None found an advantage of ω-3 over control in improving Schirmer’s values and hence tear production (both basal and stimulated).

Tear meniscus height (TMH), a non-invasive measure of tear volume, was assessed in Yilmaz’s study [41] using anterior segment OCT. The distance between the upper (corneal) and lower (lower eyelid) meniscus junctions was measured accurately using on-screen calipers. However, differences in TMH between ω-3 and control were again not found.

#### 3.2.6. Corneal Nerve Fiber Density

Corneal nerve fiber density (CNFD) decreases when there is damage to the corneal nerve plexus. This can occur post-surgery, or in the vicious cycle of dry eye and ocular surface damage [55,56].

Only one study on CXL-DED [42] measured CNFD via in vivo confocal microscopy (IVCM). As expected, CNFD decreased one month after CXL for both ω-3 and control groups, but at the 3-month mark, ω-3 treated patients had higher CNFD than controls.

#### 3.2.7. Biochemical Parameters

Inflammatory mediators such as cytokines and eicosanoids in tears can serve as markers of inflammation at the ocular surface [10]. However, only Downie’s study on CLD [44] studied these markers. The investigators collected and analyzed cytokines from basal tear samples collected at week 0 (baseline), 4, 12, and 14 [57]. They found that tear IL-17A, a cytokine mediator of inflammation, was decreased from baseline at week 12 in the topical ω-3 and oral ω-3 groups (fish oil or flaxseed oil) compared to placebo, with topical ω-3 inducing the greatest decrease in IL-17A.

#### 3.2.8. Adverse Events

The frequency of adverse events attributable to topical ω-3 was low across all five studies. Most came from the largest study of 240 DED patients by Downie 2020 [43], ranging from conjunctival hyperemia (n = 2), eye irritation/itch (n = 3), eyelid margin crusting (n = 1), and chalazion (n = 1), to foreign body sensation (n = 1). One patient experienced “colds” [44]. However, we cannot be certain whether these were related to underlying OSD or the treatment per se. No adverse events were reported in the studies by Jacobi, Yilmaz, or Cagini.

#### 3.2.9. Risk of Bias

On the whole, all studies were assessed to be either good or fair quality, signifying a generally low risk of bias, as shown in Table 4. One study [42] did not categorically specify if the treatment allocations, patients, and physician-providers were masked to the interventions, hence increasing the risk of bias.

### 3.3. Animal Studies

#### 3.3.1. Overview of Animal Studies

We included five animal studies in our review: two rabbit DED models [46,47], two murine DED models [48,49], and one canine (healthy) model [45].The method of DED induction varied among the studies, dependent on the animal used. Both rabbit studies involved dessicating eyedrops (atropine or benzylalkonium (BZA). Both mouse studies used daily subcutaneous scolpolamine injection and controlled dessicating environmental conditions of high airflow and relative humidity <30% up to a week. These methods induce aqueous-deficient DED. To better characterize ω-3 metabolism and toxicity, we included studies on healthy animals where available. The treatment and control arms also varied widely among the studies: 1% DHA-EPA vs. vehicle; 1.2%/0.85%/0.45% DHA vs. normal saline control; ALA/LA/ALA + LA vs. vehicle; ω-3 (unspecified) vs. hyaluronic acid, linseed oil (57% ω-3, 16% ω-6, 37% others). The method of DED induction, if performed, also varied among the studies. In all induced DED models, the treatment was given daily until the timepoint for a particular measurement, after which the animals were sacrificed for biochemical tissue analysis. In the healthy dog model, treatment was for 1 month and measurements were taken before and after.

**Table 3 ijms-23-13156-t003:** Human studies.

Author, Year	Condition	Fatty Acid; Regimen	Vehicle	Control	Other Tx	Outcomes Assessed and Their Results	Adverse Effects
Jacobi 2022 [40]	DED (old)	0.2% DHA qid; 8 weeks	F_6_H_8_ (surfactant)	NA (single-arm)	Lid hygiene ^	tCFS (NEI): CFB − 3.4 ± 2.1 (*p* < 0.0001) TBUT: CFB + 4.18 ± 2.77 (*p* < 0.0001) MGD score: CFB − 4.1 ± 3.3 (*p* < 0.0001) OSDI: clinically remarkable reduction by 17.5 ± 20 points (*p* < 0.0001) Slit-lamp: improved conjunctival injection (n = 14); improved lid redness (n = 2); improved PEE (n = 6); improved MGD (n = 2) NSD: Schirmer’s	None reported
Yilmaz 2021 [41]	CXL (young)	1.2 mg EPA ester + 0.02 mg DHA ester qid; 4 weeks	VE, GLY, PAA, AAC, NaOH, Na_3_PO_4_ (Remogen Omega)	HA	Moxi-floxacin qid 1 week + FML qid 4 wks	tCFS (Oxford): omega-3: 0.68 ± 0.69; HA: 1.2 ± 0.72 *p* = 0.012 TBUT (postop): omega-3: 12.56 ± 2.8; hyaluronate: 9.72 ± 3.0 *p* < 0.001 Tear meniscus(postop): omega-3: 0.38 ± 0.08 hyaluronate: 0.34 ± 0.06 *p* = 0.047 NSD: Schirmer’s	None reported
Cagini 2020 [42]	CXL (young)	DHA-EPA 1 drop tid, 3 mo	VE, carbopol 980, GLY, Pemulen, NaOH, Na_3_PO_4_ (Resolvis Omega)	HA	Ofloxacin qid 1 week + Netidex qid 2 wks + HA qid 1 mo	OSDI: benefit over HA control NF density: NF density was 6 ± 0.82 in omega-3 group and 1 ± 0.51 in control group (sodium hyaluronate) *p* = 0.0001 at 3 month follow-up	None reported
Downie 2020 [43]	DED (old)	Flaxseed oil (ALA), bid, 3 mo	CMC 0.5%, GLY 1%, P80, ECO, osmoprotectants (levocarnitine, erythritol, trehalose)	ROA	NR ^	tCFS: CFB − 1.5 ± 2.4 (*p* < 0.007 relative to control and *p* < 0.003 relative to baseline). Conj. staining: −0.85 ± 3.6 (*p* < 0.039 relative to control and *p* < 0.05 relative to baseline) NSD: TBUT, OSDI	Conjunctival hyperemia (n = 2); Eye irritation/itch (n = 3); Eyelid margin crusting (n = 1); Chalazion (n = 1); Foreign body sensation (n = 1)
Downie 2018 [44]	CLD-DED (young)	0.025% EPA + 0.0025% DHA qid, 3 mo	VE, GLY, PAA, AAC, NaOH, Na_3_PO_4_ (Remogen Omega)	Olive oil	AT	IL-17A (tear): −76.2 ± 10.8% relative to baseline (*p* < 0.05) and placebo (*p* < 0.05) NSD: corneal staining, conj. staining, TBUT, Schirmer’s, MGD score, CLDEQ-8 score, slit-lamp examination	

Data reported as mean (range) or mean ± S.D. All 5 studies were parallel in design. ^ = other concomitant treatment was not otherwise specified in the study, although the concomitant treatment(s) or lack thereof can be inferred from study-specific inclusion and exclusion criteria. Acronyms used: PO = prospective observational; RCT = randomized controlled trial; CLDEQ-8 = contact lens dry eye questionnaire-8, CLD = contact lens discomfort, CXL = collagen cross-linking; CFB = change from baseline, NSD = no significant difference between ω-3 and control; T = treatment arm; C = control arm; ω-3 = topical omega-3; EPA = eicosapentaenoic acid; DHA = Docosahexaenoic acid; ALA = alpha-linoleic acid; NSD = no significant difference between ω-3 and control; F_6_H_8_ = perfluorohexyloctane, CMC = carboxymethylcellulose, VE = vitamin E, PAA = polyacrylic acid, AAC = alkyl acrylate crosspolymer; GLY = glycerol, P80 = polysorbate 80; ECO = emulsified castor oil, HA = hyaluronic acid; FML = fluorometholone; NaOH = sodium hydroxide; Na_3_PO_4_ = sodium phosphate; qid = 4 times a day; tid = 3 times a day; bid = 2 times a day; pd = once a day; TBUT = tear breakup time; CNFD = corneal nerve fiber density; MGD = meibomian gland disease; tCFS = total corneal fluorescein staining. Scoring systems: tCFS (NEI): Jacobi (2022): scored 0–3 for each of the 5 areas of cornea (higher values indicate worse staining). Downie (2020): scored 1–5 for each of the 5 areas of cornea, and score 1–6 for each of the 6 areas of conjunctiva. tCFS (Oxford): Oxford score: 0 = absent, 1 = minimal; 2 = mild; 3 = moderate; 4 = marked; 5 = severe. Conjunctival staining: scored on 0–5 for each of the 6 zones of conjunctiva. MGD score: with Korb Meibomian Gland Evaluator on 5 central glands on lower eyelid. Each gland scored from 0 to 3, total score 15 (0 = normal; 1 = thick; 2 = paste; 3 = none (occluded)).

**Table 4 ijms-23-13156-t004:** Risk of bias (ROB2) assessment for human studies.

Author, Year	Q1	Q2	Q3	Q4	Q5	Q6	Q7	Q8	Q9	Q10	Q11	Q12	Q13	Q14	Quality Rating
Jacobi 2022 * [40]	Y	Y	Y	Y	Y	Y	Y	N	Y	Y	N	NA			Good
Yilmaz 2021 [41]	Y	Y	NR	Y	Y	Y	Y	Y	Y	Y	Y	Y	Y	Y	Good
Cagini 2020 [42]	Y	Y	NR	NR	NR	Y	Y	Y	Y	Y	Y	Y	Y	Y	Fair
Downie 2020 [43]	Y	Y	Y	Y	Y	Y	Y	Y	Y	Y	Y	Y	Y	Y	Good
Downie 2018 [44]	Y	Y	NR	N	Y	Y	Y	Y	Y	Y	Y	Y	Y	Y	Good

Y = yes, N = no, NA = not applicable to study, NR = not reported in study. The NIH Tool for Quality Assessment of Controlled Interventional Studies was used to assess all randomized controlled trials. * As this was a prospective observational cohort study, the NIH Quality Assessment Tool for Observational Cohort studies was used. For both tools, a quality rating of good, fair, or poor was given to each study based on the answers to the questions.

#### 3.3.2. Effects on Tear Function Parameters

Outcome measures on tear function parameters were sparsely and heterogeneously reported across the animal studies (Table 5); however, the methods of measuring and grading these were similar to those used in human studies. In healthy dogs, Schirmer’s was not significantly improved with respect to the fellow eye (control) or with respect to baseline [45]. In rabbits, Schirmer’s increased from 10 to 28 mm and improved TBUT after 14 days of treatment, compared to control [46]. Corneal staining was reported in two mouse studies [48,49]. In Rashid’s study, topical ALA exhibited significant decrease in corneal staining compared to control at day 10 of treatment; however, a combination of linoleic acid (ω-6 PUFA) and ALA failed to show improvement compared to control. Another study [49] reported no significant superiority of either concentration of ω-3 used (0.2% or 0.02%) over HA control; however, ω-3 combined with HA topical treatment was superior to HA alone.

#### 3.3.3. Effects on Cytokine Elaboration and Immune Cells

Effects of topical ω-3 on conjunctival cytokine elaboration were reported in three studies [45,48,49]. All animal studies [45,46,47,48,49] took conjunctival biopsies, from which one study [49] used multiplex immunobead assay to quantify cytokine proteins and two other studies [45,48] used real-time polymerase chain reaction to assess the mRNA levels of each cytokine gene.

Across these three studies [45,48,49], two mouse studies [48,49] reported a beneficial effect of topical ω-3 in reducing inflammatory cytokine elaboration. Intriguingly, in these two studies, the effects on cytokines tended to follow the same trend as the corneal staining. In Rashid’s mouse study, treatment with topical ALA decreased corneal and conjunctival expression of IL-1A and TNF-a at day 10 compared to vehicle control, but a combination of LA and ALA failed to show improvement compared to control. Similarly, in Li’s study, either concentration of ω-3 failed to reduce IL-1b and IL-17 compared to HA control, but the combined ω-3 with HA topical treatment demonstrated significant reduction in IL-1B and IL-17 compared to HA control. The last study on healthy dogs [45] assessed mRNA levels of these cytokines in both conjunctival biopsy and tears. The study found no significant difference in inflammatory cytokines (Table 5) between ω-3-treated and control groups. Since healthy dogs are not expected to be deficient in ω-3 to start with (compared to induced DED animals), the lack of response to topical ω-3 in this regard is not surprising.

Effects on immune cells were assessed in two studies [47,48]. In a mouse DED model [48], cell counts of CD11b+ monocytes, which are responsible for antigen presentation and induction of innate immune response, were quantified from freshly excised corneas by incubation with CD11 antibodies and counting the number of positively-stained cells in each zone (arbitrarily defined) under confocal microscopy. They found that CD11b+ monocytes in the center cornea significantly decreased in the ALA-only group compared to vehicle control ALA-LA, and LA-only groups (*p* = 0.03), but a similar trend was not found for the peripheral cornea (*p* = 0.07). However, a rabbit DED model [47] showed no differences in goblet cell density in the conjunctival biopsy between linseed oil (57% ω-3; 16% ω-6; 37% other fatty acids) and placebo groups. While linseed oil has a beneficial ω-3:ω-6 ratio and is theoretically anti-inflammatory, the actual percentage concentrations of these PUFAs in the formulation used were not reported and could very well be much lower than those used in the other studies.

## 4. Discussion

From the limited evidence in this focused systematic review, we attempted to assess the efficacy of topical ω-3 in the treatment of ocular surface inflammatory disease.

In terms of tear film function parameters, across all studies (human and animal) and all DED subtypes (idiopathic, CXL-DED or CLD-DED), topical ω-3 improved corneal staining in the majority of human [40,41,43] and animal studies [48,49] that reported this. Dryness symptoms [40,42] and tear stability (TBUT) [40,41,46] were also improved. There was no statistically significant benefit of topical ω-3 on Schirmer’s test in the human studies [40,41,44] and animal studies [45,46] that assessed it.

However, there were differences in outcomes between chronic (DED/CLD) and iatrogenic (CXL) dry eye. Among the chronic dry eye studies [40,43,44], one-third reported improved TBUT, one-third reported improved symptom score, and half reported improved MGD score. Most of the equivocal results came from the CLD study [44], possibly due to milder baseline signs and symptoms and relatively lower concentration of ω-3 used. Among the CXL studies [41,42], there was improvement in signs (tCFS, TBUT, TMH, OSDI, and CNFD) and symptoms (OSDI). However, each outcome was assessed only by one study, which is insufficient.

Regarding biochemical parameters, IL-17 was reduced in ω-3 treated patients compared to control in two studies that assessed it. There is insufficient evidence (reported by only one study [48]) to make definitive conclusions on CNFD, CD11b+ cell enumeration, and IL-1b elaboration, but these were improved in w3-treated patients compared to controls.

The ratio of ω-6 to ω-3 might be important. In Rashid’s mouse model, a mixture of 0.1% LA and 0.1% ALA delivered worse results on corneal staining and inflammatory cytokines compared to 0.2% ALA alone, possibly due to an increase in the ω-6:ω-3 ratio in tear film post-treatment and hence a lesser anti-inflammatory effect. However, the studies did not measure or account for the baseline ω-3 levels or ω-3:ω-6 ratio among all subjects (human/animal). Not all DED patients are deficient in dietary ω-3, hence not all will respond to supplementation (be it topical or oral). No studies directly compared effects of oral versus topical ω-3.

While the anti-inflammatory and pro-resolution effects of ω-3 metabolites are well-established, their molecular mechanisms of action on the ocular surface are less well-elucidated. Mechanisms of ω-3 improving MGD and conjunctivitis have been studied. Incubation of human meibomian gland epithelial cells with DHA led to a sustained decrease in COX-2, and pro-inflammatory cytokines such as IFN-y, IL-6, TNF-a compared to controls [58]. Pre-treatment of human conjunctival cells with DHA before artificially inducing inflammation decreased expression and secretion of pro-inflammatory proteins eotaxin-1, and RANTES, compared to vehicle control. Eotaxin-1 is a chemokine implicated in recruitment of eosinophils in allergic inflammation [59]; RANTES is involved in homing and migration of CD8 T cells [60].

Topical ω-3 (in particular, DHA) might also promote regeneration of the corneal nerve plexus, which is damaged in DED, as demonstrated in Cagini’s study in our review and is borne out elsewhere [61,62]. One explanation is that human corneal epithelial cells express 15-LOX, which converts topical DHA into neuroprotectin D1 (NPD1) [63], which is understood to be a pro-resolving mediator [64,65]. After corneal epithelial injury, PEDF increases and upregulates 15-LOX expression [66]. Furthermore, PEDF acts synergistically with DHA to cause corneal wound healing, as reviewed separately [67]. Taken together, these studies suggest that corneal nerve injury in the vicious cycle of ocular surface inflammation and tear film dysfunction can, at least to some extent, be reversed by topical ω-3. However, although 15-LOX is largely pro-resolving [68], other metabolites may be pro-inflammatory, as explored in a separate review [69].

Effects of topical application of ω-3 metabolites have also been studied. Resolvin E1 has shown strong anti-inflammatory activity in mouse models of induced DED [37,38], showing superior effects on goblet cell preservation, maintenance of tear secretion, and reduced CD11b+ monocytes and CD3+ T cell (antigen presenting cell) infiltration into the cornea.

Despite the idea of ω-3 given directly to the ocular surface as a topical formulation being mooted as early as 2008 in a mouse model of induced DED [48], uptake of this idea has been lukewarm at best, with the first study of topical ω-3 in humans only performed 10 years later by Downie in the treatment of contact lens discomfort, as found in our review. This might be due to existing barriers to adequate drug delivery to this location [70]. Eyedrops instilled into the tear film undergo rapid pre-corneal clearance due to blinking and reflex tearing, leading to immediate overflow and drainage of the excess [34]. Furthermore, ω-3 PUFAs, which are lipophilic, do not mix well in aqueous eyedrop formulations. Many methods are being theorized and investigated to increase retention duration of ω-3 at ocular surface, such as viscosity enhancers, colloidal nanocarriers, liposomes, micelles, and surfactants [71]. For example, Novaliq’s EyeSol^®^ technology uses amphipathic molecules such as perfluorohexyloctane that dissolve lipophilic compounds and have long-lasting interactions with the tear film lipid layer, increasing residual time of the drug in the eye [40]. Such advances in pharmaceutical sciences help address, at least in part, the aforementioned challenges of delivering and retaining ω-3 at the ocular surface.

In spite of such challenges, it is worthwhile optimizing and exploring formulations of topical ω-3, since the incorporation of topical ω-3 into the therapeutic armamentarium for ocular surface disease confers many advantages. As a supplement and not a pharmacological compound, ω-3 is suitable for use in the community without the need for specialist referral. Second, topical ω-3 can be used in mild ocular surface inflammation (e.g., DED) or even prophylaxis in prolonged screen users, given its favorable side effect profile, as elucidated through our review. Further large-scale trials on topical ω-3 based on the promising results in our review might be able to propel w3-containing eyedrops into the treatment paradigm for ocular surface diseases such as DED. Third, since ω-3 can be incorporated into pre-existing artificial tear formulations which patients will be taking anyway, compliance will be less of an issue compared to taking separate oral ω-3 capsules.

However, the findings of our study must be interpreted in the context of known limitations. The most important limitation is the lack of measurement of tear ω-3 at baseline. Since not all DED cases have deficient ω-3 in tears, further supplementation of ω-3 to tears may be useless in these patients. This is in accordance with Wilder’s principle: the pre-treatment level determines the response to that treatment [72]. Secondly, the DED induced in these animals may not represent human physiology. For example, the rabbit models that used lacrimal gland removal or topical atropine/BZA induced aqueous-deficient but not evaporative DED [73]. However, by the time most DED patients present, they have a combination of both evaporative and aqueous-deficient DED [15]. Third, not all studies masked the outcome assessors to the assigned intervention. In studies investigating treatments for DED, masking of the investigator is crucial as the measurements of certain outcomes such as TBUT or tCFS may be subjective; prior knowledge of a participant’s intervention may bias the way the investigator interprets and reports the findings. Of the studies included in our review, only three [41,43,44] out of five human studies and two [48,49] out of five animal studies masked the assessors to the participants’ assigned intervention.

Another important limitation is the heterogeneity among all studies, which limited the extent to which results could be consolidated. First, the etiology of dry eye among the studies varied (CLD, CXL, idiopathic DED) and hence differences in response could very well be due to differences in underlying pathophysiology. For instance, iatrogenic DED by nature has fewer elements of chronicity and conjunctival fibrosis [74]. Efforts at resolving inflammation, regardless of effectiveness, cannot reverse fibrosis. Second, the composition, ratios (if a combination was used), and concentration of ω-3 used varied among the studies, which may influence the overall anti-inflammatory effect because of differences in relative concentrations of the DHA, EPA, and ALA metabolites. Third, not all outcomes were reported by all studies, hence the results for a particular outcome may be biased in favor of those particular studies that reported them. Furthermore, since it was difficult to objectively pool the results as different scoring systems were used (Oxford vs. various modifications to NEI), comparing the number of studies supporting a particular hypothesis may suffer from sampling bias. Fourth, some studies excluded patients with conditions which are found fairly commonly among the dry eye population, such as allergic conjunctivitis, regular anti-glaucoma eyedrop use, and conjunctivochalasis, hence potentially decreasing the applicability of their results. Last, the paucity of evidence in this area prohibits establishment of any firm conclusions.

To address the aforementioned limitations, future trials should consider assessing tear eicosanoid levels (prostaglandins, leukotrienes, neuroprotectins, resolvins, etc.) among all studies pre- and post-treatment. Different doses of the ω-3 used should be used to assess dose-dependent effects and hence optimize dosing. Different types of ω-3 (EPA, DHA, ALA) should also be compared with each other and in combination.

## 5. Conclusions

In conclusion, topical ω-3 is a promising treatment option for DED and other ocular surface inflammatory disease. In humans, there was promising evidence for improved ocular surface staining and TBUT compared to controls; equivocal evidence for OSDI and MGD, no effect on Schirmer’s I or II tests, and promising data on restoring CNFD. Data from animal models largely agreed with these findings and revealed decreases in inflammatory cytokines IL-17 and infiltration of CD11b+ monocytes. Further trials in human are required to more accurately characterize effects as well as optimize dosage and combination of ω-3. Basic science studies aimed at elucidating mechanisms of and devising drug delivery methods for ω-3 will be helpful.

## Figures and Tables

**Figure 1 ijms-23-13156-f001:**
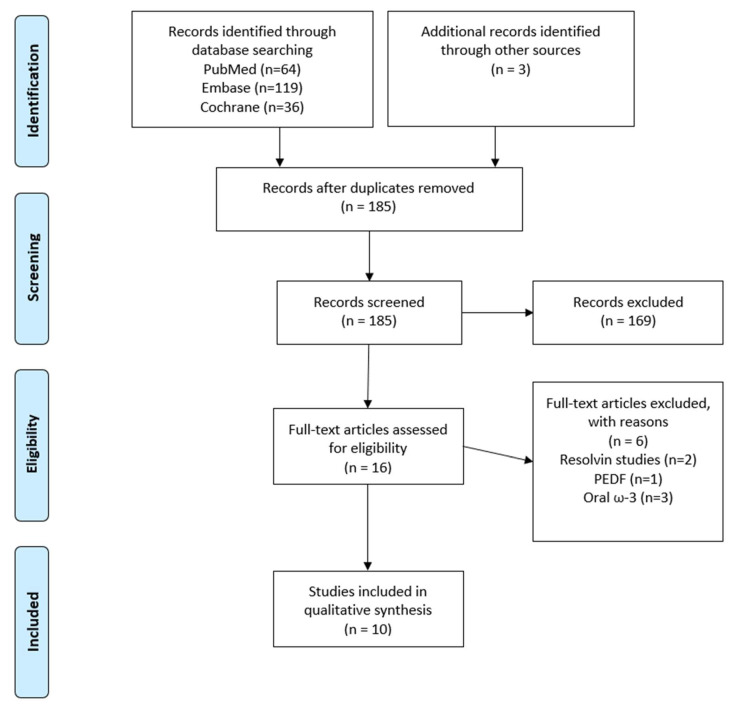
PRISMA flow diagram (PEDF = pigment epithelial-derived factor).

**Table 1 ijms-23-13156-t001:** Summary of human studies.

Author, Year	Sample Size	Study Type, Masking	Condition	Gender	Age Range	Significant Inclusion Criteria	Significant Exclusion Criteria
Jacobi 2022 [40]	T: 33	PO, open-label	DED (old)	T: 69.7%	T: 54.8 ± 17.86	History of DED at least 6 mos, tCFS (NEI) < 11, MGD > 3, TBUT < 8, OSDI > 25, Schirmer’s I > 5 mm Pts must have applied lid hygiene for 14 days prior to and continue during study	Oral medications causing ocular drying Contact lenses Clinically significant slit-lamp findings requiring prescriptive treatment Topical eyedrops within 30 days Lipid eyedrops within 15 days Procedures involving meibomian gland in past 6 months Scarring, irradiation, alkali burns, cicatricial pemphigoid, destruction of conjunctiva, abnormal lid anatomy
Yilmaz 2021 [41]	T: 25 C: 25	RCT, double-masked	CXL (young)	T: 24% C: 36%	T: 24.24 ± 2.46 C: 24.64 ± 2.22	Progressive keratoconus	Corneal thickness < 400 μm Corneal hydrops/scar tissue Conjunctivochalasis Previous ocular surgery Systemic medications that affect tear film Uncontrolled diabetes/hypertension/liver/kidney Contact lenses within past 3 months Signs of persistent ocular surface changes Topical steroids, cyclosporine Moderate-severe DED before operation Allergic conjunctivitis Perioperative complications from CXL
Cagini 2020 [42]	T: 20 C: 20	RCT, NR	CXL (young)	T + C: 45%	T + C: 28 (22–37)	Stage 1 keratoconus No signs of ocular surface damage	NR
Downie 2020 [43]	T: 120 C: 120	RCT, double-masked	DED (old)	T: 70% C: 77.9%	T: 54.3 ± 17.3 C: 52.8 ± 16.7	Aged > 18, with DED, OSDI > 18 and <65 at baseline	Schirmer’s < 2 mm tCFS > 18 at baseline Systemic medications that affect tear film Ocular surgery or trauma affecting corneal sensation or tear distribution Contact lens use Glaucoma eyedrops
Downie 2018 [44]	T_1_: 14 T_2_: 14 T_3_: 14 C: 14 #	RCT, single-masked	CLD (young)	T_1_: 86% T_2_: 59% T_3_: 83% C: 75%	T_1_: 25.9 ± 1.9 T_2_: 29.4 ± 1.3 T_3_: 23.3 ± 0.6 C: 24.6 ± 1.3	Soft disposable contact lens use (>5 days/week, for past 6 months) CLDEQ-8 score ≥ 13 (symptomatic CLD) No change to topical/systemic medications affecting tear film status	Active ocular infection, inflammation, allergy Severe corneal disease (excluding superficial punctate keratitis), severe blepharitis Use of topical steroids, cyclosporine, NSAIDs Systemic medications affecting tear film Punctal plugs History of ocular surgery

Data reported as mean (range) or mean ± S.D. All 5 studies were parallel in design. # quadruple arm: oral EPA + DHA vs. oral EPA + DHA + ALA vs. topical EPA + DHA vs. olive oil control. T_1_: topical ω, T_2_: oral fish oil; T_3_: oral fish + flaxseed oil; C: oral placebo. Acronyms used: PO = prospective observational; RCT = randomized controlled trial; CLDEQ-8 = contact lens dry eye questionnaire-8, CLD = contact lens discomfort, CXL = collagen cross-linking; CFB = change from baseline, NSD = no significant difference between ω-3 and control; T = treatment arm; C = control arm.

**Table 2 ijms-23-13156-t002:** Summary of animal studies.

Author, Year	Animal, Model	Description of Study
Damani 2014 [45]	Dogs, healthy	This study investigates the outcomes of a topical w-3 formulation of DHA and ALA applied to the eye of healthy dogs, with their fellow eye used as control. Slit-lamp examination, ocular surface fluorescein staining, and tear cytokine levels (both protein and mRNA) were measured from Schirmer’s test strips.
Lidich 2018 [46]	Rabbit, induced DED	This study describes the preparation of various microemulsions containing riboflavin 5-phosphate (RFP) and various other surfactants, to be mixed with triglyceride DHA (TG-DHA), to be used in ex vivo and in vivo experiments on a rabbit model of induced DED. Ex vivo experimentation sought to determine the effect of RFP on biomechanical strength of the cornea. In vivo experimentation aimed to determine the effect of TG-DHA on tear breakup time and Schirmer’s test.
Neves 2013 [47]	Rabbit, induced DED	This study investigates efficacy of linseed oil given orally, topical, and oral–topical combined, on ocular surface staining with fluorescein and rose bengal, and on Schirmer’s test. Histopathologic analysis was also performed to evaluate conjunctival goblet cell density.
Rashid 2008 [48]	Mouse, induced DED	This study evaluates effectiveness of various topical w-3 formulations of ALA and LA. Corneal fluorescein staining was used to assess the integrity of the ocular surface pre- and post-treatment. Immunohistochemistry with microscopy was used to evaluate CD11b+ cells in various regions of the cornea. Corneal and conjunctival tissues were separately homogenized and the mRNA levels of various inflammatory cytokines were quantified with RT-PCR and qPCR.
Li 2014 [49]	Mouse, induced DED	This study evaluates the efficacy of various mixtures of w-3 with hyaluronic acid applied topically. Corneal fluorescein staining was used to assess the integrity of the ocular surface pre- and post-treatment. Conjunctival tissues were homogenized and immunobead assays were used to quantify concentrations of inflammatory cytokines IL-1b, IL-17, and IP-10, as well as lipid peroxidation markers hexanoyl-lys and 4-hydroxynonenal.

Abbreviations: DHA = docosahexaenoic acid, ALA = alpha-linoleic acid, DED = dry eye disease, IL = interleukin, RT-PCR = reverse transcriptase polymerase chain reaction, qPCR = quantitative polymerase chain reaction.

**Table 5 ijms-23-13156-t005:** Animal studies and results.

Author, Year	Animal, Model	Method of DE Induction	Fatty Acid and Regimen	Control	Vehicle	Outcomes Assessed and Their Results
Damani 2014 [45]	Dogs, healthy	N/A	1% DHA + 1% EPA tid, 1 month	Vehicle	None specified	NSD: Schirmer’s, cytokines (IFN-y, TNFa, IL-1a, IL-1b, IL-2, IL-6, IL-8, and IL-10 (analysis from tears and conjunctival biopsy)
Lidich 2018 [46]	Rabbit, induced DED	BZA	1.2, 0.85, 0.45 wt% DHA @ ^	NS	T80, Cremophor EL	TBUT: at 14 days: ω-3 > 10 s; control = 5–7 s (no statistical analysis done) Schirmer’s: at 14 days: ω-3 = 28 mm; control = NA (no statistical analysis)
Neves 2013 [47]	Rabbit, induced DED	Lacrimal gland Removal + topical atropine	linseed oil (57% ω-3, 16% ω-6) bid @	Placebo	NR	NSD: conjunctival goblet cell density
Rashid 2008 [48]	Mouse, induced DED	s/c scopolamine + wind + low humidity	0.2% ALA, 0.1% ALA + 0.1% LA, 0.2% LA 1 x/day @	Vehicle	T80, Glucam e-20, VE, packing solution	Staining: decreased (compared to control) at days 5, 10 Cells: CD11b+ cells in the center of cornea significantly decreased (*p* = 0.03) in ALA-only group compared to vehicle control Cytokines: ALA-only group showed decreased corneal and conjunctival expression of IL-1A and TNF-a at day 10, compared to vehicle control
Li 2014 [49]	Mouse, induced DED	s/c scopolamine + wind + low humidity	ω-3 qid @	HA	NR	Staining: 0.2% ω-3 + HA = 4.35 ± 1.40; significantly higher compared to 0.2% ω-3 and 0.02% ω-3 + HA. Cytokines: 0.2% ω-3 + HA: Conjunctival IL-1b, IL-17 significantly lower compared to HA only, 0.02% ω-3 only, 0.2% ω-3 only, and 0.02% ω-3 + HA.

@ until timepoint studied. ^ dosage per day not reported. ω-3 = topical omega-3; EPA = eicosapentaenoic acid; DHA = docosahexaenoic acid; ALA = alpha-linoleic acid; NSD = no significant difference between ω-3 and control; BZA = benzylalkonium chloride, NS = 0.9 wt% sodium chloride solution; packing solution = water, boric acid, NaCl, EDTA; T80 = tween-80; qid = 4 times a day; tid = 3 times a day; bid = 2 times a day; pd = once a day.

## Data Availability

The data presented in this study may be available on request from the correspondence author.

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
