# Peer review of "Topical Omega-3 Fatty Acids Eyedrops in the Treatment of Dry Eye and Ocular Surface Disease: A Systematic Review"

_ijms, 2022, doi:10.3390/ijms232113156_

Round 1
Reviewer 1 Report
The authors present a systematic review approaching the evidence of topical ω-3 in dry eye disease. The review is well researched and contains the most important studies regarding topical ω-3 treatment in human and animal studies. In my opinion, the choice to perform a descriptive review over a meta-analysis is adequate due to the heterogeneity of the available data. Nevertheless, just because of the heterogeneity of the available studies, I would suggest – for the purpose of better readability of the review – to summarize the included studies separately. Otherwise, the text is very difficult to read.
Minor recommendations:
- Line 32: „reason“
- Line 34: insert „the“
- Line 37: I suggest mentioning more precise studies for specific reasons of DED.
- Line 41: Please cite some studies, which indicate abnormalities in tear film quantity and quality.
- Line 205: Add “Gland”
- Line 230: remove “it”
- Line 204: replace “1” by “One”
- I suggest the authors to indicate significant results throughout the manuscript.
- Line 320-322: This sentence should be further specified, otherwise it appears confusing.
- In my opinion, in studies investigating treatments for DED, masking of the investigator is crucial, as the investigation methods may be very subjective – for example the investigation of BUT or corneal staining. I would suggest including this to the discussion.
- Conclusion: I would prefer to use the term “treatment option”.
Reviewer 2 Report
This is a good systematic review. Congratulations. I have been impressed because of the fact that authors have considered humans and animals. However, the conclusions are correctly written and there, authors give a response to the aim of this work.
I have many minor suggestions:
Introduction:
- Line 54: “ω -3 (ω-3) fatty acids”, probably it is easier to write: ω -3 fatty acids (ω-3) and Line 58: ”ω -3 fatty acids (ω -3 PUFAs)”, ω -3 fatty acids (ω -3) or Line 59: “omega-6 (ω-6) PUFAs “ as omega-6 (ω-6)
- Line 66, Line 310, Line 336: w-6:w-3, use ω
- Line 74: GIT, what does it mean?
Methodology:
In my opinion, all the text must be written in an impersonal stile.
- Line 96: (CXL) must be written in line 96 and not in line 139.
Results:
- Line 147: It seems to be a double space.
- Line 195: “…by tear film break-up time (TBUT), which is the time interval elapsed between a complete blink and the first appearance of a break in the tear film” it might be written in 119 line.
- Line 210: Probably, it could be easier to read if 5 was changed by five.
- Line 211: You have to include bibliography.
- Line 251: “all studies were assessed to be either good or fair quality”. This sentence is good without the bibliography, but when bibliography is consider, it is nor clear. Change this sentence, please.
- Line 255: Probably, it could be good to speak about the different among animals DED. Even, if there are different between animals and humans DED. All that information could be included in "Overview of animal studies" or in the introduction.
- Line 275: What w.r.t. means?
- Line 277: “in 2 mouse studies”, Line 284: “ in 3 studies”, Line 285: “All studies”, “1 study used”, Line 286: “2 other studies”, Line 288: ” Across these three studies”, Line 301: “Effects on immune cells were assessed in 2 studies”: Insert bibliography
Discussion:
- Line 327, Line 333, Line 339, Line 390: w-3 might be ω-3
- Line 332: Insert bibliography
- Line 378: Novaliq’s “EyeSol”, must be written EyeSol®
- Line 407: DE is DED
Figure 1:
It could be better to include the records from each date base used.
What do you mean with "resolvin studies"?
What is PEGF? Each figure and table must be understood by it self.
Write correctly "Oral w-3"
Round 2
Reviewer 1 Report
The authors have adressed all my suggestions adequately. I do not have any more comments.